# Mesopelagic fishes dominate otolith record of past two millennia in the Santa Barbara Basin

William A. Jones[1] & David M. Checkley, Jr. [1]*

The mesopelagic (200–1000 m) separates the productive upper ocean from the deep ocean, yet little is known of its long-term dynamics despite recent research that suggests fishes of this zone likely dominate global fish biomass and contribute to the downward flux of carbon. Here we show that mesopelagic fishes dominate the otolith (ear bone) record in anoxic sediment layers of the Santa Barbara Basin over the past two millennia. Among these mesopelagic fishes, otoliths from families Bathylagidae (deep-sea smelts) and Myctophidae (lanternfish) are most abundant. Otolith deposition rate fluctuates at decadal to centennial time scales and covaries with proxies for upper ocean temperature, consistent with climate forcing. Moreover, otolith deposition rate and proxies for temperature and primary productivity show contemporaneous discontinuities during the Medieval Climate Anomaly and Little Ice Age. Mesopelagic fishes may serve as proxies for future climatic influence at those depths including effects on the carbon cycle.

[1] Scripps Institution of Oceanography, University of California, San Diego, CA 92093–0218, USA. *email: dcheckley@ucsd.edu

Ocean margins are the most productive seas due to the large supply of nutrients by upwelling, mixing and rivers that maintains high levels of primary productivity and in turn support ecologically and economically valuable fisheries[1]. In the California Current System, fluctuations of fish populations, including collapses, have profound effects on fisheries and ecosystems, such as starvation of charismatic megafauna like the California sea lion[2]. The relative roles of fishing and the environment in collapses are uncertain but important to understand for fisheries management[3].

To date, the majority of data regarding fluctuations of fish populations comes from commercially important epipelagic (0–200 m) species[4]. Yet beneath the sunlit epipelagic, the mesopelagic contains fishes that are either permanent residents or migrate to the sea surface each night to feed, with implications for ecosystem functioning and energy fluxes[5]. Indeed, the abundance of these mesopelagic fishes has recently been revised upward by an order of magnitude[6] and they have been shown to contribute to biodiversity and the downward flux of carbon[7]. Moreover, the mesopelagic often coincides with oxygen minimum zones (OMZs), which are projected to expand under climate change[8]. Yet despite the importance of the mesopelagic to ocean ecology, biodiversity and the carbon cycle, little is known of its long-term variation. Here, we hypothesize the occurrence of otoliths in layers of anoxic sediments might provide insight into the abundance and fluctuations of dominant types of pelagic, mesopelagic and demersal fishes in the overlying water column over the past two millennia.

Otoliths are structures of calcium carbonate and protein in the semi-circular canals of bony fishes and used to sense acceleration and orientation. Bony fishes contain three pairs of otoliths, the sagittae being the largest and the lapilli and asterisci smaller[9]. The size and shape of otoliths vary with species and age. Sagittal otoliths of adult fishes common in the Santa Barbara Basin (SBB) have been classified to taxonomic group based on size, shape and elemental composition (see Methods section).

The SBB has a maximum depth of 600 m and lies southeast of Point Conception in Southern California (Fig. 1). Sediments in cores from the SBB have been dated[10] and analyzed for planktonic diatoms and silicoflagellates[11], foraminifera[12] and fish scales[13–15], enabling reconstruction of time series of proxies for upper ocean temperature and primary production[16] and fish abundance[13–15] with decadal resolution over the past two millennia.

We show that mesopelagic fishes dominate the otolith assemblage of the Santa Barbara Basin over the past two millennia. Fishes of two families, Bathylagidae and Myctophidae, contribute the most otoliths and their rates of deposition vary similarly to proxies for upper ocean temperature and primary productivity and hence climate. Otoliths of mesopelagic fish in the sediment may be a proxy for the effects of climate at these depths.

## Results and discussion

**Recovery and classification of fossil otoliths.** A total of 1524 otoliths were recovered from four cores collected from the central SBB. Three Kasten cores (Supplementary Fig. 1) dating from 8–53 A.D. to 1885–1924 A.D. contained 454, 438, and 549 otoliths. One box core dating from 1836 to 2004 contained 83 otoliths. Of the 1524 total otoliths, 336 (22%) were too altered to classify. Of the remaining 1188 otoliths, expert opinion was used to classify 1013 otoliths (85%) into five families (Supplementary Fig. 2); 175 (15%) were unable to be identified and were classified as 'Other'. Otoliths were from, in order of numerical abundance (percent of identified otoliths), Myctophidae (lanternfish, 41%), Bathylagidae (deep-sea smelts, 36%), Merlucciidae (hake, 11%), Engraulidae (anchovy, 7.6%), and Sebastidae (rockfish, 4.4%). Classification of otoliths by expert opinion was consistent with their classification based on morphological and elemental features and feature-based classifiers developed using otoliths of known origin (see Methods section, Supplementary Table 1). Hereafter, we focus on Bathylagidae, Myctophidae and Composite (all identified) otoliths classified by expert opinion.

Mesopelagic fishes (Bathylagidae, Myctophidae) were the dominant (77% of identified) source of otoliths in the SBB sediments over the past two millennia (Fig. 2, Table 1). Dominance of the SBB otolith record by mesopelagic fishes is consistent with recent upward revisions, by an order of magnitude, of their biomass locally and globally based on scientific acoustics and net sampling[6]. Mesopelagic fish otoliths were abundant in surface sediments of the Western Mediterranean[17] and myctophid otoliths were abundant in the surface sediments of the NW Atlantic[18] and Pleistocene sediments in the eastern Mediterranean[19]. Our results for fossil otoliths are consistent with recent observations in the SBB. The bathylagid *Leuroglossus stilbius* and myctophid *Stenobrachius leucopsarus* comprised 90% of fishes captured from the upper 500 m[20] and 68% of fishes captured in the upper 70 m[21] by midwater trawl in the SBB. *Leuroglossus stilbius* was abundant in in situ visual observations to 542 m in the SBB[22].

**Reconciling otolith and scale records.** The fish scale record for the past two millennia in the SBB is dominated by northern anchovy (*Engraulis mordax*), Pacific sardine (*Sardinops sagax*) and Pacific hake (*Merluccius productus*)[13–15]. Dominance of the scale record by these taxa can be reconciled with mesopelagic fishes dominance of the otolith record in the SBB by considering the taxon-specific production and fate of scales and otoliths. Anchovy, sardine and hake produce numerous, large and robust scales that preserve well[23]. Anchovy and sardine shed scales when attacked[24], enhancing their scale production. Mesopelagic fishes dominant in trawl collections from the SBB either do not have scales (the bathylagid *L. stilbius*)[25] or have fewer scales that are very thin and thus unlikely to preserve well (the myctophid *S. leucopsarus*)[26]. Dominance of anchovy, sardine and hake in the SBB scale record is consistent with these observations. Unlike the variable numbers and productivity of scales, all bony fishes contain a fixed number of three pairs of otoliths, the sagittae

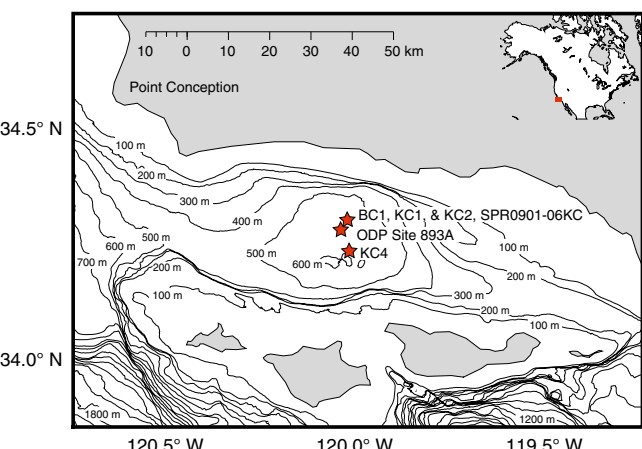

**Fig. 1** Map of Santa Barbara Basin showing locations of sediment cores. BC1, KC1, and KC2 in this study were taken at station MV1012-ST46.9 (34° 17.228′ N, 120° 02.135′ W). KC4 was taken at station MV1012-ST46.2 (34° 13.700′ N, 120° 01.898′ W). Cores from SPR0901–06KC (34° 16.914′ N, 120° 02.419′ W) and ODP Site 89 A (34° 17.25′ N, 120° 02.2′ W) were used to establish chronology used in this study (refs. [10,42,43])

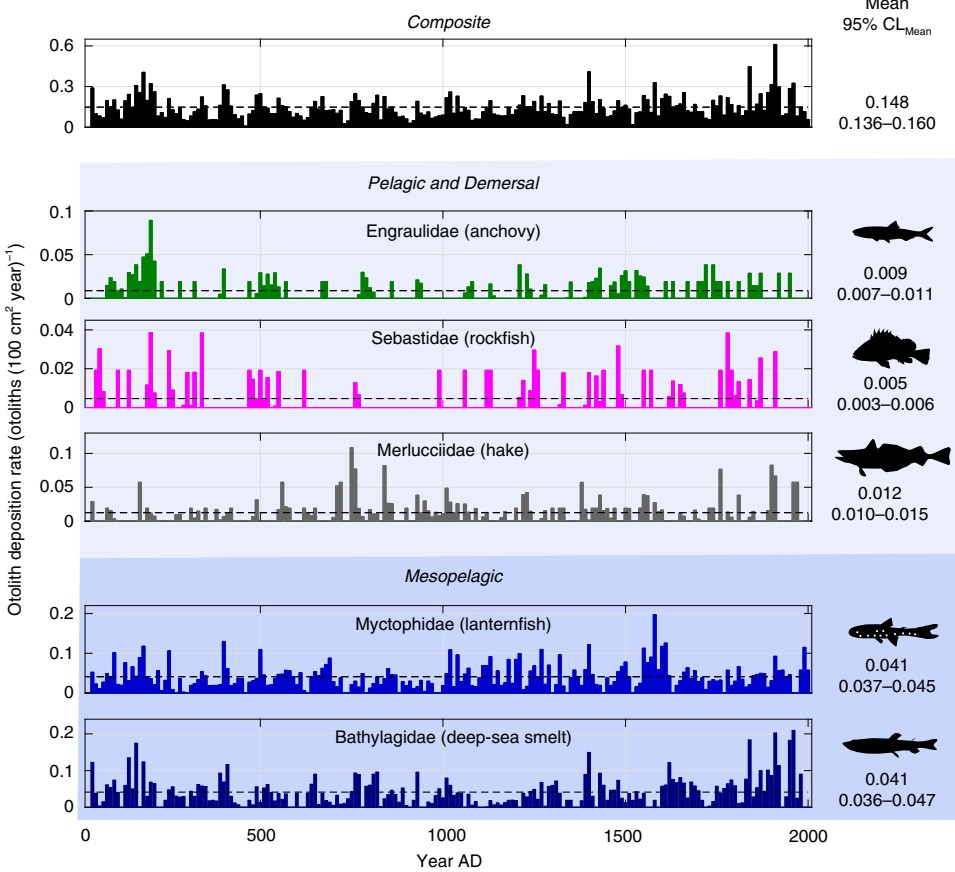

**Fig. 2** Otolith deposition rates (ODR) for all fishes and the five dominant families. Vertical bars represent 10-year averages of ODR (otoliths (100 cm$^2$ year)$^{-1}$). Mean ODR shown as horizontal dashed lines and on right with 95% confidence limit of mean ($n_{bins} = 197$). Composite includes otoliths from all five families and unidentified (Other): $n_{otoliths} = 1188$. Engraulidae: $n_{otoliths} = 77$. Sebastidae: $n_{otoliths} = 46$. Merlucciidae: $n_{otoliths} = 110$. Myctophidae: $n_{otoliths} = 413$. Bathylagidae: $n_{otoliths} = 367$. Identification by expert opinion (see Methods section). Shading indicates upper ocean (light blue) and mesopelagic (dark blue). Source data are provided as a Source Data file

**Table 1 Otolith classification by fish habitat**

| Classification | $n_{otoliths}$ | Percentage of classified otoliths | Percentage of identified otoliths |
|---|---|---|---|
| Pelagic | 77 | 6.5 | 7.6 |
| Demersal | 156 | 13.1 | 15.4 |
| Mesopelagic | 780 | 65.7 | 77.0 |
| Other | 175 | 14.7 | |
| Total | 1188 | 100 | 100 |

Pelagic are Engraulidae. Demersal are Merlucciidae and Sebastidae. Mesopelagic are Bathylagidae and Myctophidae. Other are otoliths unable to be classified to family. Otoliths from all cores (KC1, KC2, KC4, and BC1) have been combined. Highly altered otoliths have been excluded ($n_{otoliths} = 336$) (see Methods section). Source data are provided as a Source Data file

being the largest[9]. Otolith preservation among taxa, however, is poorly known but is likely related to otolith size, favoring anchovy, sardine and hake (large otoliths) over mesopelagic fishes including Bathylagidae and Myctophidae (small otoliths) (see Methods section). *Cyclothone* spp. (Gonostomatidae), while globally abundant in the mesopelagic, were not observed in the SBB otolith record, consistent with their rarity in SBB trawl samples[20] and the small size of their otoliths[27]. Anchovy, sardine and hake spawn primarily outside the SBB[28] and thus are transient in the SBB and it is unlikely all die there. Mesopelagic fishes are believed to be permanent residents in the SBB[20] and it is likely they die there. Mesopelagic fishes dominance of the SBB otolith

record, despite the relatively small size of their otoliths, is consistent with their dominance in trawl surveys[20,21]. The otolith and scale records are complementary and provide valuable insights into the occurrence of anchovy, sardine and hake (scales) and mesopelagic fishes (otoliths) over the past two millennia in the SBB.

**Otolith deposition rate and the environment**. Otolith deposition rate (ODR, otoliths (100 cm$^2$ year)$^{-1}$) did not show a significant trend over two millennia for individual families or Composite (Fig. 2), consistent with an absence of otolith alteration in the sediment. Discontinuities in ODR time series occurred early (~200–300 A.D.) and later (~1600–1800 A.D.) (Fig. 3). The latter discontinuity corresponds to the end of the Little Ice Age (LIA)[29]. Bathylagidae and Myctophidae, in addition, showed complementary but opposite discontinuities ~1000–1100 A.D., near the beginning of the Medieval Climate Anomaly (MCA)[29]. ODR has increased since 1850 AD by 50% (0.14 to 0.21), 62% of which was due to Bathylagidae (0.038–0.082, 117% increase) (Fig. 2).

Time series of proxies of sea surface temperature (SST) and primary productivity (PROD), based on the $\delta^{18}O$ of planktonic foraminifera in sediment cores from the SBB[16], were compared with ODR (Fig. 3) in the time domain. We used only environmental proxies based on sediment cores from the SBB to maximize comparability with ODR. SST and PROD were inversely correlated (Kendall's $\tau = -0.40$, $p < 0.0001$, $n = 188$)

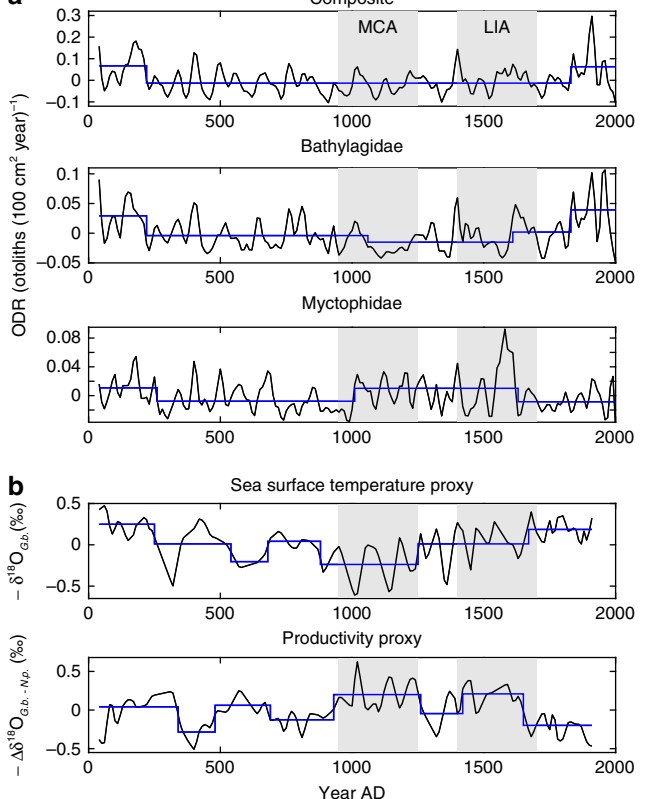

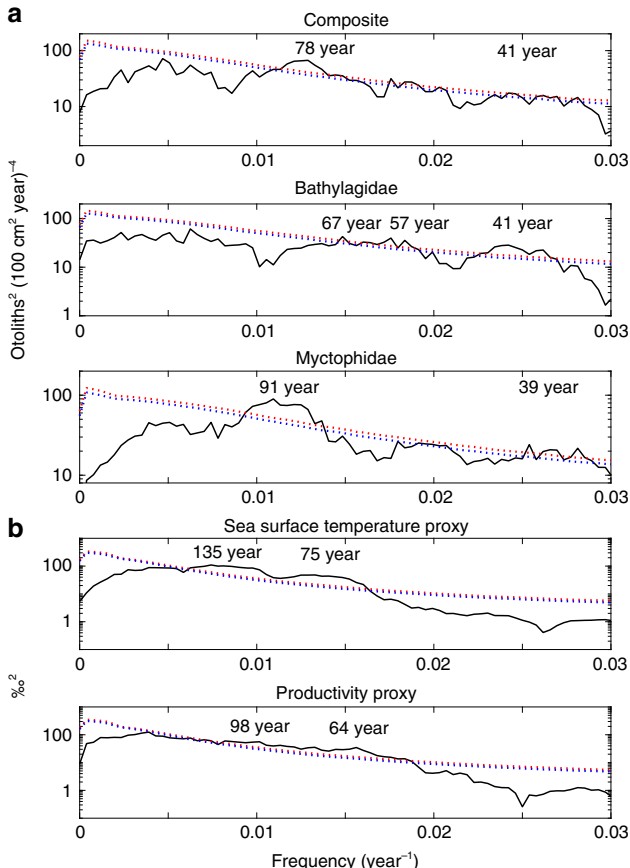

**Fig. 3** Otolith deposition rate and environmental proxies. **a** Otolith deposition rate (ODR) for Composite (all classified otoliths, $n_{otoliths} = 1188$), Bathylagidae ($n_{otoliths} = 367$) and Myctophidae ($n_{otoliths} = 413$). 10-year binned ODR time series ($n_{bins} = 197$) were detrended. Resulting anomaly time series were averaged using a 4-bin (40-years) double-running mean (black lines, see Methods section). **b** Proxies for sea surface temperature and primary productivity (10-year bins, $n_{bins} = 188$, black lines). Blue lines are results from Sequential $t$ Test Analysis of Regime Shifts (Methods) with a cutoff length, $l$, of 20-bins (200 years) and a significance value, $p$, of 0.05 (two-tailed). Discontinuities indicated by vertical blue lines. Mean values indicated by horizontal blue lines. Gray indicates Medieval Climate Anomaly (MCA) and Little Ice Age (LIA) (ref. [29]). Source data are provided as a Source Data file

**Fig. 4** Global power spectra for otolith deposition rate and environmental proxies. **a** Global power spectra for otolith deposition rate of Composite (all classified otoliths, $n_{otoliths} = 1188$), Bathylagidae ($n_{otoliths} = 367$), and Myctophidae ($n_{otoliths} = 413$). **b** Global power spectra for proxies of sea surface temperature and primary productivity (10-year bins, $n_{bins} = 188$). Filtered to remove periods less than 30 years and greater than 500 years. Dotted lines are confidence limits (red, 95%; blue, 90%; two-sided) computed assuming a red-noise background with first-order autocorrelation. Periods of major peaks exceeding 95% are labeled. See Methods section for details. Source data are provided as a Source Data file

and their time series were dome-shaped over the two millennia, with SST minimal and PROD maximal ~1000 A.D. (Fig. 3). Multiple discontinuities were found for SST and PROD (Fig. 3). SST was low and PROD high during the MCA, while SST was high during the LIA.

Climate forcing of mesopelagic fishes would be manifest by similar variation of ODR and the environment in the frequency domain. ODR (Bathylagidae, Myctophidae), SST and PROD had discontinuities near the start and end of the MCA and end of the LIA (Fig. 3). Global power spectra for time series filtered to remove periods less than 30 years, due to uncertainty in core chronology, and greater than 500 years, due to limits of 2000 years time series, showed peaks in ODR variability at periods of 39–91 years and in SST and PROD at 64–135 years (Fig. 4). Similar peaks were observed for unfiltered time series (Supplementary Fig. 3). Wavelet spectra for ODR showed peaks at periods of 60–393 years and for SST and PROD at 83–197 years but not stationary (Supplementary Fig. 4). Variation in periods of significant peaks in spectra of fish abundance and climate proxies is attributed, in part, to the data coming from different cores. Long-term relationships between SST and the abundance of

midwater taxa are evident from comparisons of cumulative summations of SST and Composite ODR ($r = 0.72$) and SST and Bathylagidae ODR ($r = 0.71$) (Fig. 5). Collectively, analyses of time series of ODR for mesopelagic taxa and Composite with SST and PROD indicate similar variation over a range of scales, from multidecadal to centennial. The combined results in the time and frequency domains provide strong inference that climate affects mesopelagic fish.

SST and PROD in the SBB and other regions of the ocean vary with climate, including upwelling-favorable winds which bring cool, nutrient-rich water to the upper, sunlit euphotic zone. Climate may also affect ocean circulation, as reflected in the nutrient content of upwelled water[30] and variation in OMZs[31]. Epipelagic conditions affect mesopelagic fishes in at least three ways. First, the larval stages of most mesopelagic fishes live in the epipelagic, feeding on zooplankton[32]. Second, migrating meso-pelagic fishes feed nightly on near-surface zooplankton[33]. Third, the downward flux of particulate matter nourishes the mesope-lagic ecosystem, including its fishes[34]. Our results indicate that mesopelagic fishes have dominated the fish assemblage repre-sented in the otolith record of the SBB and varied with SST and PROD, and thus climate, over the past two millennia. Debate exists over the cause of climate variability over the range of scales

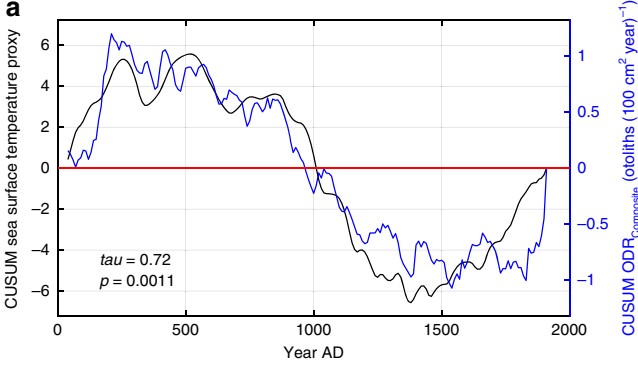

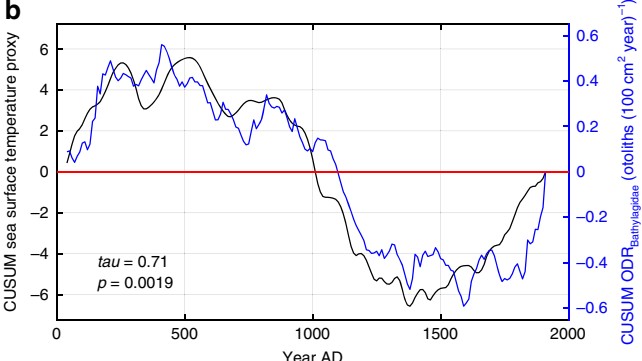

**Fig. 5** Cumulative summations of otolith deposition rate and sea surface temperature proxy. **a** Sea surface temperature (SST) proxy and Composite (all classified otoliths) otolith deposition rate (ODR). **b** SST proxy and Bathylagidae ODR. ODR time series detrended before cumulative summation (CUSUM) analysis. Kendall's *tau* and associated probability (*p*) shown. Probabilities computed using distribution of *tau* from time series with distributional and autoregressive properties of ODR time series. Time period is 40–1910 A.D. with 10-year resolution and $n_{bins} = 188$. Red line is zero CUSUM. Source data are provided as a Source Data file

observed in this study. Multidecadal, particularly 50–70 years but also shorter and longer, periods are characteristic of the Pacific Decadal Oscillation[35] and have been observed for scale deposition rate for anchovy and sardine in the SBB[14]. Longer periods, to 260 years, are characteristic of solar activity cycles[36] and have been observed for plankton and the oxygen minimum zone in the nearby Gulf of California[37] and the deposition rate of combined anchovy and sardine scales in the SBB[38].

**Mesopelagic fishes and climate**. The SBB mesopelagic fish assemblage is relatively large and low diversity[20]. While our results are specific to the SBB, the conclusion that the mesopelagic fish assemblage varies with climate may be general. Linkage to climate may be modulated at least in part by epipelagic primary production, which nourishes the mesopelagic through the active and passive transport of matter and energy[34,39]. Indeed, members of the world's most abundant vertebrate, *Cyclothone* spp. (Gonotomatidae), are permanent residents of the mesopelagic and have recently been shown to rely on particles sinking from the epipelagic[40]. Climate-related variation of the epipelagic thus affects the mesopelagic. To our knowledge, our SBB otolith record is the first documentation of long-term natural variability of mesopelagic biota and its relation to climate. Such environmental sensitivity indicates ecosystem services provided by the mesopelagic, including fisheries, biodiversity and the biological pump[7], may vary with climate variability and change. We have shown for the SBB that mesopelagic fishes dominate the otolith assemblage

### Table 2 Core types, locations and water depths

| Core | Type | Latitude | Longitude | Water depth |
|------|------|----------|-----------|-------------|
| BC1 | Box | 34° 17.228′ N | 120° 02.135′ W | 580 m |
| KC1 | Kasten | | | |
| KC2 | Kasten | | | |
| KC4 | Kasten | 34° 13.700′ N | 120° 01.898′ W | 600 m |

in sediments, consistent with their high abundance and likely importance in the carbon cycle. Further, our results show climate effects may influence mesopelagic ecology. Finally, otoliths of mesopelagic fishes may serve as proxies of the influence of a changing climate at those depths.

## Methods

**Cores and chronology**. Three Kasten cores (KC1, KC2, and KC4) and one box core (BC1) were collected from near the center of the Santa Barbara Basin (SSB) (Fig. 1) in 2010 (Table 2)[41].

Color photographs of each core were taken on the deck of the ship before subcoring. The three Kasten cores and one box core were subcored on deck with rectangular acrylic core liners ~76-cm long and 15 cm by 15 cm in cross section. Each Kasten core yielded four end-to-end subcores and the box core one subcore. All subcores, in acrylic liners, were placed into Hybar trilaminate membrane bags with oxygen absorbers, flushed with nitrogen, vacuum-sealed, and stored at 4 °C. This storage method was successful in maintaining anoxic conditions within the sediments for several months until sample processing. One vertical slab ~2 cm thick was trimmed off the side of each subcore and X-radiographed at the Scripps Institution of Oceanography Geological Collections using a Geotek MSCL-XR core scanner. The core slabs were scanned at 1-mm intervals in a linear, non-rotational scan. Individual two-dimension images were combined to make the composite X-radiograph images.

X-radiographs and color photographs were used to develop a high-resolution chronology for each core. Several age models have been developed to assign dates to the SBB varved stratigraphy. The traditional age model relied on counting seasonal varve couplets[42] and was used to establish a chronology for the top ~35 cm of BC1 in the present study. Analysis of [14]C dates from planktonic foraminiferal carbonate and terrestrial-derived organic carbon from Kasten core SPR0901–06KC showed that accuracy of the traditional varve counting method decreased prior to ~1700 A.D. due to under-counting of varves[10,43]. Using [14]C dates, a new SBB chronology was established from ~107 B.C. to 1700 A.D.[10,43]. We used this new SBB chronology to develop our Kasten core chronology, as follows. First, the major, near-instantaneous sedimentation events characterized and dated in Kasten core SPR0901–06KC were identified in our cores and assigned a single calendar date (Supplementary Fig. 5)[10,42,43]. The overall varve structure of each core was then cross-dated between our Kasten cores and core SPR0901–06KC to aid in the identification of layers formed rapidly during flood or turbidite events (Supplementary Fig. 1)[10]. The down-core chronology of each Kasten core was then corrected by excluding these near-instantaneous events. Finally, for each Kasten core, a series of linear regression equations between sequential, near-instantaneous events were developed to assign dates to the remaining stratigraphic structure[10].

Varves of BC1 were counted from 2009 back to 1871 A.D. Visual cross-dating agreed well with the chronology of box cores dated previously[42]. The sediment chronology prior to 1871 for BC1 and for each entire Kasten core was resolved every 0.5 cm, exclusive of near-instantaneous event layers, by cross-dating as described above[10,43], extending the chronology of BC1 back to 1836 A.D. and the Kasten cores back to 8–53 A.D. The bottom ~50 cm of KC4 was not processed due to low confidence in the stratigraphic chronology (Supplementary Fig. 5).

Each 15-cm × 13-cm cross section subcore was cut transversely every 0.5 cm, if no near-instantaneous event layer occurred, to create transverse sections (97.5 cm³). Near-instantaneous event layers were added to respective transverse sections, in which cases the volume exceeded 97.5 cm². Transverse sections were stored frozen at −80 °C until further processing. Each transverse section represented a time interval of ~2 to 8 years. The date assigned each transverse section was the average of the dates of its upper and lower surfaces.

**Otolith removal from sediment**. Each transverse section was thawed, dried overnight at 50 °C, rinsed in distilled water and wet-sieved using a 104-μm mesh. Otoliths in the >104-μm-mesh fraction of sediment in each transverse section were found visually using a dissecting microscope, removed manually using forceps, stored dry individually in plastic vials and assigned unique numbers (otolith_id).

**Otolith preservation**. A single experiment on the preservation of otoliths of fish species common in the otolith record of the SBB was conducted as part of an unpublished senior thesis (Mark Morales, University of California San Diego, Bachelor of Science, 2014). Sagittal otoliths of *Engraulis mordax* (*n* = 12), *Sardinops*

**Table 3 Otolith dissolution rate estimates**

| Family | Area of recent[a] otoliths (mm$^2$) ($\bar{x} \pm se(n_{otoliths})$) | Dissolution rate of recent[a] otoliths ($\mu^2$ min$^{-1}$) | Area of fossil[b] otoliths (mm$^2$) ($\bar{x} \pm se(n_{otoliths})$) | Dissolution rate of fossil[b] otoliths ($\mu^2$ min$^{-1}$) |
|---|---|---|---|---|
| Engraulidae | 3.5 ± 0.2 (82) | 2.8 | 1.8 ± 0.2 (93) | 4.7 |
| Clupeidae | 1.0 ± 0.1 (10) | 6.1 | n/a[c] | n/a[c] |
| Merlucciidae | 21 ± 3 (19) | very low[d] | 3.9 ± 0.6 (130) | 2.4 |
| Sebastidae | 1.1 ± 0.1 (24) | 6.0 | 0.83 ± 0.14 (48) | 6.4 |
| Bathylagidae | 0.90 ± 0.02 (181) | 6.2 | 0.62 ± 0.02 (422) | 6.8 |
| Myctophidae | 1.2 ± 0.1 (368) | 5.6 | 0.890 ± 0.04 (413) | 6.3 |

Source data are provided as a Source Data file
[a] recent otoliths of known species, one (left or right sagitta) per fish (ref. [47])
[b] fossil otoliths from this study
[c] no fossil otoliths classified as Clupeidae
[d] otolith area outside range of regression

*sagax* ($n = 11$), *Sebastes* spp. ($n = 11$), *Bathylagus wesethi* ($n = 9$), and *Stenobrachius leupcopsarus* ($n = 8$) were incubated at pH 2 and room temperature for up to 4.5 h. Dissolution rate (DR, rate of change of otolith area, $\mu m^2$ min$^{-1}$) was negatively related to initial otolith area (OA, mm$^2$): DR = $8.26e^{-0.310\ OA}$ ($n = 51$, $r^2 = 0.73$, $p < 0.001$). We used this equation to estimate the DR of otoliths of known (recent) and classified (fossil) family (Table 3).

Fossil otoliths from SBB sediments are of unknown origin with respect to species and path from live fishes to the sediment. Otoliths may or may not have passed through the digestive tract of a predator. Predators (e.g., squid, fishes, marine mammals and seabirds) vary in regard to digestive tract conditions (pH, temperature, gut passage time)[44]. Thus, we are unable to correct for otolith degradation from fish death to otolith recovery from the sediment. Larger otoliths (e.g., recent Engraulidae, Clupeidae, and Merlucciidae) preserved better than smaller otoliths (e.g., recent Bathylagidae, Myctophidae) in this preliminary experiment, consistent with otolith recovery in marine mammal feces[45,46]. Dominance of Bathylagidae and Myctophidae in the fossil otolith record of the SBB is consistent with their dominance as a source in the overlying fish assemblage.

*Cyclothone* spp. otoliths were not observed in the fossil otolith record of the SBB (present study). *Cyclothone* spp. otoliths are small[27] (~0.2 mm$^2$) and thus their dissolution rate is predicted to be high. The absence of *Cyclothone* spp. in the fossil otolith record is consistent with the rarity of this genus in trawl collections from the SBB[20] and the likely poor preservation of their otoliths.

**Otolith deposition rate**. Transverse Section Otolith Deposition Rate (TSODR, otoliths (100 cm$^2$ year)$^{-1}$) was first calculated for each transverse section for each core from the number of otoliths in a section (195 cm$^2$), normalizing to 100 cm$^2$, and dividing by the section duration in years. Otolith Deposition Rate (ODR, otoliths (100 cm$^2$ year)$^{-1}$) for each 10-year bin for each core was calculated by summing the TSODR(s) spanning and/or included in each 10-year bin, apportioning TSODR spanning a bin limit by the proportion of the duration of a transverse section in that 10-year bin. Finally, the average ODR for all cores was calculated for each 10-y time bin. Otoliths accumulated at different rates for the three Kasten cores, less so when expressed as a fraction of total otoliths in a core (Supplementary Fig. 6).

**Otolith analysis**. Details of methods of otolith analysis are published[47]. A summary follows.

Otoliths recovered from sediments were subjectively assigned an index of otolith alteration, ranging from 2 (least altered) to 10 (most altered)[41,45]. Otoliths with an otolith alteration score 8–10 were classified but excluded from further analysis. Otoliths with an otolith alteration score 2–7 were classified and used in further analyses.

A color image of each otolith was created using a microscope with reflected light and a black background. Otolith images were converted to gray scale and then to binary images, in which the otolith (white) was differentiated from the background (black) using a threshold of pixel intensity. Twelve geometric (GEO) features and a boundary contour were extracted. The boundary contour was expressed as chain-coded points, and then approximated by 120 elliptical Fourier (EF) descriptors. The EF descriptors were normalized and combined into x and y components, resulting in 29 x and 30 y EF features.

A subset of fossil otoliths (Supplementary Table 1) was randomly selected from the pool of all fossil otoliths (Supplementary Table 1), cleaned, and analyzed for elemental composition using solution-based mass spectrometry with a Thermo Finnigan Element2 single collector inductively coupled plasma mass spectrometer (ICP-MS). The elements $^7$Li, $^{23}$Na, $^{25}$Mg, $^{39}$K, $^{55}$Mn, $^{88}$Sr, and $^{138}$Ba were measured and expressed as a ratio with respect to measured $^{48}$Ca[48]. The seven element ratios comprised the element features (ELM).

**Otolith classification**. Otoliths were classified into one of seven groups: the families Bathylagidae, Clupeidae, Engraulidae, Merlucciidae, Myctophidae, and Sebastidae, which represent the most common taxa found in the Santa Barbara Basin (SBB) region[49], and Other. Otoliths classified as Bathylagidae include those missing the rostrum (Broken Bathylagidae, Supplementary Table 1). Classification was performed by experts and by models using morphological and element features[47]. Summaries of these methods are provided below.

Expert opinion (Expert) classification consisted of two experts independently classifying each fossil otolith by visual comparison with recent otoliths from fishes of known species[41,47]. If the two experts classified the same otolith differently in the first round, each expert repeated the visual classification. If disagreement remained, the otolith in question was classified as Other.

Two classification models developed previously[47] were used to classify fossil otoliths. Discriminant Function Analysis (DFA) and Random Forest Analysis (RFA) were used. Both models were trained with modern otoliths from fishes common in the southern California Current System and SBB regions and utilized only the 10 strongest discriminatory features[47,50]. RFA10 Rank uses ten morphological features (9 GEO, 1 EF) selected using a procedure based on rank of occurrence[47]. DFA10 SW uses 8 morphological and 2 element features (7 GEO, 1 EF, and 2 ELM) selected using a stepwise procedure[47]. RFA10 was used to classify all fossil otoliths. DFA10 SW was used to classify the subset of fossil otoliths analyzed for elemental composition.

A classification was developed using DFA and expert classification results which utilized all available types of data, i.e., expert, morphology and elemental composition. Using this method, an otolith was assigned to a taxonomic-based class or Other only when the classification of Expert and DFA10 SW agreed.

ODR for all families (Bathylagidae, Clupeidae, Engraulidae, Merlucciidae, Myctophidae, and Sebastidae) and Other were summed to create the Composite class.

**Climate proxies**. We compared the SBB ODRs with two proxies of the environment. We restricted the environmental time series to having been derived from comparable sediment cores from the SBB. Time series of proxies for sea surface temperature (SST) and productivity (PROD) were provided by Dr. James Kennett (University of California, Santa Barbara). Both proxies are based on measurements of oxygen isotopes ($\delta^{18}$O) in shells of fossil foraminifera recovered from SBB sediments[51]. SST proxy is based on the negative relationship of water temperature to the oxygen isotopic composition of the surface-dwelling, planktonic foraminifer *Globigerina bulloides* ($\delta^{18}$O$_{G.\ bulloides}$)[51]. Our SST proxy is $-\delta^{18}$O$_{G.\ bulloides}$; higher values indicate warmer water, lower values cooler water. PROD proxy is based on the negative relationship of the difference between the isotopic compositions of *G. bulloides* and the deeper-dwelling *Neogloboquadrina pachyderma* ($\Delta\delta^{18}$O$_{G.\ bulloides-N.\ pachyderma}$) and water column stratification[16,51]. Our PROD proxy is $-\Delta\delta^{18}$O$_{G.\ bulloides-N.\ pachyderma}$; higher values indicate higher primary productivity, lower values indicate lower primary productivity. The chronology of the SST and PROD proxies was updated using the most recent SBB chronology[10,52], enabling direct comparison with ODR time series. Proxy data were resampled at 10-year intervals and detrended before further analysis.

**Statistical analyses**. Central tendency was characterized by the mean (Matlab *mean.m*, Mathworks). $n$ was the number of observations. Variability was characterized by standard deviation of samples (Matlab *std.m*, Mathworks) and standard error of the mean and coefficient of variation[53].

Kendall rank-order non-parametric correlation procedure was used to test for correlation between times series of pairs of variables[54]. The Kendall–Mann procedure was used to test for long-term trends in time series[54,55].

Autocorrelation was measured for use in creating red noise time series to evaluate significance of power spectra and wavelet analysis results. Input time series were first detrended (Matlab *detrend.m*, Mathworks). Autocorrelation was then computed (Matlab *crosscorr.m*, Mathworks).

Global power spectra were used to test for periodicity in time series. The Thomson multitaper method (Matlab *pmtm.m*, Mathworks) was used on

detrended (Matlab *detrend.m*, Mathworks) input ODR time series. Significance of the spectral peaks was assessed by comparing spectral peaks with a red-noise spectrum[56]. The red-noise spectrum was created by making 1000 random time series with the same length, mean, standard deviation and first autocorrelation coefficient (AR(1)) as the input time series. Values were ranked and probability distributions created for each frequency, allowing assessment of spectral peak significance at a specified level (e.g., 95%). Global power spectra were created both on unfiltered time series and after band-pass filtering the input time series (Matlab *filtfilt.m*, Mathworks).

Morlet wavelet analysis was used to test for non-stationary periodicity in time series[57]. A wavelet analysis characterizes the spectral components of a time series as a function of time, using a moving window, thus allowing examination of time dependence of different periodic components of a time series[57]. The input ODR time series was detrended and then normalized by subtracting the mean and dividing by the standard deviation. The wavelet transform was then applied (Matlab *wavelet.m*, Mathworks). Significance of wavelet peaks was assessed by their comparison with a red-noise distribution with the same AR(1) as the input time series. Values below the 'cone of influence' are considered 'dubious'[57].

Cumulative sum (CUSUM) analysis was used to test for long-term coherence in pairs of time series[58]. CUSUM was calculated by summing the detrended ODR from early to late (Matlab *cumsum.m*, Mathworks). A negative slope in the CUSUM plot identifies a period in the time series where values are below the long-term mean, while a positive slope identifies a period of in the time series where values are above the long-term mean. A horizontal line represents periods near the mean[58]. Kendall's non-parametric, rank-based correlation *tau* was computed for pairs of CUSUM time series, i.e., one ODR and either SST or PROD[55]. The significance of Kendall's *tau* was evaluated by comparison with probability distributions of Kendall's *tau* created as follows[59]. The AR(1) of the ODR input time series was estimated and used to create a distribution of random values with the same mean and standard deviation. Kendall's *tau* was computed 10,000 times for the randomized ODR and the observed SST or PROD CUSUM time series. The results were used to create a probability distribution of Kendall's *tau* for each combination of ODR and either SST or PROD CUSUM time series. This distribution was used to assign a probability to observed Kendall's *tau* for each combination of ODR and either SST or PROD CUSUM time series.

CUSUM was also used to test for differences between rates of accumulation of otoliths in the three Kasten cores. The Kolmogorov-Smirnov test (Matlab *kstest2.m*, Mathworks) was used to test for differences in the rate of accumulation of otoliths between pairs of Kasten cores.

The Sequential *t* Test Analysis for Regime Shifts (STARS) was used to test for discontinuities between periods of relative constancy in a time series[60]. The input time series were detrended (Matlab *detrend.m*, MathWorks). Composite, Bathylagidae and Myctophidae ODR input time series were smoothed using a double-running mean (Matlab *filtfilt.m*, Mathworks) using four 10-year bins. STARS parameters were 20 bins (200 years) for the cutoff length (*l*) and 0.05 for two-tailed probability level (*p*)[60].

**Reporting summary**. Further information on research design is available in the Nature Research Reporting Summary linked to this article.

## Data availability

The fossil otolith and environmental proxy data that support the findings of this study are available in UC San Diego Digital Collections (http://doi.org/10.6075/J0154FC9) (ref. [61]). The source data underlying Figs. 2, 3, 4, and 5, Tables 1 and 3, Supplementary Figs. 3, 4, and 6 and Supplementary Table 1 are provided as a Source Data file.

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

## Acknowledgements

W.J. was supported by a National Science Foundation Graduate Research Fellowship and California Sea Grant (NOAA Award NA14OAR4170075 CHECKLEY to D.C.). We thank James Kennett for environmental proxy data, Mark Morales for laboratory help and dissolution experiment results, HJ Walker and Ben Frable of the Scripps Institution of Oceanography Marine Vertebrates Collection for specimens and Bryan Black for statistical advice and constructive comments.

## Author contributions

W.J. and D.C. conceived the study; W.J. collected the samples and data; W.J. and D.C. developed and implemented the analyses; D.C. with W.J. wrote the paper.

## Competing interests

The authors declare no competing interests.

## Additional information

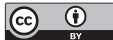

