## [Peer Review File · Nature Communications]

Reviewers' Comments:

Reviewer #1:

Remarks to the Author:

Substantive comments:

Line 17-18: should say "Bathylagidae and Myctophidae dominated the otolith record." It is not correct or appropriate to imply that the otolith record is a quantitative measure of assemblage composition. The otolith record should only be used as a RELATIVE representation of abundance over time for only those fishes whose otoliths are amenable to preservation. For example, stomiiform fishes have very small otoliths that are likely underrepresented in the record. These fishes are the numerically dominant mesopelagic fishes in the World Ocean, and presumably were so during the timespan of this scale record.

18-20: The meaning of these statements is unclear – in a global sense, gelatinous zooplankton dominate in colder waters and copepods in warmer waters. If this situation is reversed on the SBB, then this would severely limit the global application of these findings.

142-149: This section, which is one of the central foci of this paper, has major problems. The statement "The Bathylagid *L. stiblius* prefers gelatinous zooplankton, which filter feed on phytoplankton, including small cells characteristic of warm, oligotrophic conditions," is in error on many counts. First, the 'gelatinous zooplankton' span several phyla, only one of which (Chordata, subphylum Tunicata) filter feeds small phytoplankton. The majority of gelatinous zooplankton are predaceous. Second, it has not been proven that this bathylagid consumes only salps. The exact diet composition of gelatinous-feeding fishes is notoriously difficult to determine (see review papers of Arai). Third, as general (global) rule, salps tend to dominate the zooplankton of high-nutrient oceanic regimes (esp. higher latitudes), whereas crustacean zooplankton (copepods) dominate lower-nutrient oceanic regimes (lower latitudes) – the authors have stated just the opposite. Thus, this thesis in the paper may be conceptually flawed, with the following points supporting this assertion: 1) the alternate stable states suggested here are unfounded - a large portion of the mesopelagic fish taxa are not represented in this otolith record [e.g. Stomiiforms, Aulopiformes, Stephanoberyciformes, which collectively largely outnumber myctophids and bathylagids combined]; 2) the salp/copepod dynamic is grossly oversimplified – variation in the relative abundances of these taxa in the Southern Ocean are highly volatile; 3) if, for the sake of argument, the salp/copepod dynamic were completely accurate as depicted here, it would really suggest that the SBB is an aberrant marginal sea that is a poor analog for the World Ocean; and 4) the thesis presented here implicitly assumes that the SBB is a closed basin with little/no exchange between the Pacific proper, - what if the fish assemblages were continuously refreshed/homogenized by extraneous larval recruitment?

Notes on paragraph beginning in Line 151, listed by 'implication':

1. "this is the first documentation of long-term variability of mesopelagic biota..." YES – this is the most important aspect of this paper;
2. "climate appears to affect the mesopelagic realm, which acts as a filter for carbon from surface productive zone sequestered in the deep ocean." YES, another very important contribution of this paper, though, again, the mesopelagic zone is in no way a 'filter.'
3. "it is imperative to understand the dynamics of mesopelagic fish stocks, including their natural and human-caused variability, to sustainably manage their future fisheries." This statement falls under the heading, "Blinding glimpse of the obvious." It is not necessary, and is not a result from this paper. It can be deleted.
4. "Finally, two human activities, greenhouse gas emissions and fishing, have the potential to affect the mesopelagic biota." YES/NO. Yes to the first – it is a restatement of point number 2, so is essentially superfluous. No to the second – this paper did not investigate the effects of fishing on the

mesopelagic biota, so does not belong here.

General comments:

The clarity of this manuscript could be much improved. The writing is choppy and discordant, with a tendency toward stream-of-consciousness (esp. Introduction). The number of technical errors (typos, misspellings, factual errors) suggest this paper was not thoroughly reviewed and proofed before submission. Supportive examples of this assessment are listed below.

Throughout ms: "fishes" should be used in all cases when referring to more than one species.

Line 17, 46: No, not all fishes have three pairs of otoliths (e.g., Chondrichthyes).

31: the mesopelagic zone does not act as a filter. This needs to be rewritten.

76: bony

83-86: fish family common names are not capitalized; cannot start sentence with genus abbreviation.

88: formula typographical error.

89: poor grammar – what does this sentence mean?

97-99: this statement is vague to the point of being irrelevant.

129: larval

Final recommendation: this paper contains valuable information, but also several sections that require revision. I recommend the authors revise and resubmit, as the paper is not publishable as-is.

Signed: Tracey T. Sutton

Reviewer #2:

Remarks to the Author:

The central premise of this MS is excellent, and very appropriate for the readership of Nature. Fish otoliths provide an excellent means of species identification for some taxa, and tend to be better preserved in non-acidic sediments than are scales. Therefore the approach used in this study offers great potential. In general, the study seems to have been carefully done and well written, with one serious caveat: why are the results so different than those previously published for the same location? The previous studies by Soutar and Isaacs (1969) and Baumgartner et al (1982) used the same general approach, in the same location, using scales rather than otoliths. Although the authors' explanations for differential preservation of scales vs otoliths are valid, they do not provide a convincing case for the validity of one millennial reconstruction over the other. Therefore, I recommend that the MS be re-evaluated after a better 'reality check' is provided. These and other suggestions are detailed below.

Line 38: This sentence needs to refer specifically to the study location, since it implies (incorrectly) that sardines and anchovy scales occur in all anoxic sediments, and that they are the only species whose scales accumulate.

Line 72: Although it is fair to state that some fish do not have scales, it is misleading to suggest that the mesopelagic fish reported in this study do not have scales. As a matter of fact, myctophid scales are very large, are they not?

Line 78: The statement that degradation rate differences between scales and otoliths might explain the very different species composition apparent in this study compared to earlier scale-based studies is a reasonable one. However, I was surprised that there appears to have been no adjustment made for species-specific otolith degradation rates. It is well established that small otoliths or those with a high surface area to volume ratio dissolve or degrade more quickly than do more robust otoliths. As a

result, the species with more fragile otoliths tend to be under-represented in studies such as these (i.e. Tollit et al 1997. Species and size differences in the digestion of otoliths and beaks: implications for estimates of pinniped diet composition. *Can. J. Fish. Aquat. Sci.* 54:105-119). That may explain the relative absence of sardine and anchovy otoliths in this study compared to past scale-based studies reporting high abundance of sardines and anchovies in the same Basin, but does not indicate that the current study provides the more accurate species reconstruction. Therefore, I strongly suggest that steps be taken to provide a reality check for the current results. The easiest way to do this would be to relate recent fish surveys of the Basin area with the otolith-based species composition inferred from the most recent period of the sediment cores, using all species, not just the summary reported in the text. Indeed, given that the authors reported no temporal trends in species composition, perhaps the entire sediment core time series could be used. Either way, it is important to demonstrate that the two species composition patterns (survey and otolith-based) correspond.

Given that the sediment cores reflect a seasonal integration of fish abundance, it would be best if seasonal fish surveys of the Basin were used in the otolith-fish survey comparison. However, one survey is better than nothing. I note as well that one of the two references used in support of recent high abundance of midwater fishes was from the Santa Barbara Channel, not the Santa Barbara Basin. Are these different areas?

If some species seem to be under-represented in the sediment cores relative to fish surveys, then some sort of calibration will be required to adjust for the species detection bias in the cores.

Line 107: A lot of emphasis and space is given to the power spectra, without much justification for why it is done. There is very little in the way of convincing long-term periodicity evident in any fish species abundance time series, so why would you expect it here? More importantly, where are some of the other, more convincing, statistical analyses? Where is a PCA to highlight similarities among time series? Or better yet, a mixed effects model to test for significant effects of environmental proxies on the fish abundance time series?

Line 117: This analysis hinges on the assumption of a stable otolith degradation rate (not to be confused with deposition rate). Are there any studies which support the assumption that the anoxic conditions and pH at the bottom of the Basin remain stable across years, independent of SST or productivity? If so, these should be cited. If not, the implications of the bias need to be mentioned.

Line 133: The glaring omission here, both in the text and in the Results, are the pelagic fish species, particularly sardines and saurys. Why weren't any of their otoliths found? And why were so few mesopelagic fish scales reported in the earlier studies of Soutar and Isaacs and Baumgartner? And even if some explanation is found for the huge discrepancy between the authors' results and those of scale-based studies, why is there no discussion of it (other than the brief mention in the introductory comments on Line 72)? This definitely needs to be addressed in the text. Either the previous studies were hugely biased, or the current study is hugely biased. Both approaches have identifiable sources of bias. It is important that the authors convince the reader that their study is the one that should be believed.

Line 156: Although it is reasonable to assume that greenhouse gas emissions and fishing have the potential to affect mesopelagic fish abundance, there is nothing in this study that would support that conclusion.

Line 424: According to Line 398, only a subset of otoliths were analyzed for elemental composition. However, it states here that the model DFA10SW (of which two of the input variables are elemental composition) was used to identify all otoliths. One of these statements appears to be wrong.

Steve Campana

Reviewer #3:

Remarks to the Author:

This paper reports data on the otoliths of mesopelagic fish in the Santa Barbara Basin to raise several interesting points. First, mesopelagic fish dominated the otolith assemblage in the sediments, testifying to their high abundance and likely importance in carbon cycling. Second, climatic effects may influence ecological dynamics at mesopelagic depths. Third, and related, the otoliths of mesopelagic fishes should provide clues to those impacts and potentially serve as proxies for climatic influence at those depths.

There is one potentially fatal flaw in this paper. From my reading, all otoliths of all fishes were identified and enumerated (to the extent possible, of course). The issue is the danger of pseudoreplication. It is just like collecting clam shells on the beach and counting both left and right valves. Paleontologists routinely choose one or the other to avoid pseudoreplication, and studies have shown that one valve or the other can in some situations be favored to wash up depending on hydrography. Likewise, at least potentially, for otoliths: is it possible that the three types of otoliths are not equally preserved? Could there be interspecific differences in preservation? The authors should only be counting the largest otoliths on one side: the left or right sagittae, because the sagittae are the largest ones. It should be possible to rerun the analysis with only one of the sagittae.

Replies to Reviewers' Comments

Reviewers' comments are in bold. Authors' replies are in normal font. Line numbers in replies refer to revised manuscript.

Reviewer #1 (Remarks to the Author):

Substantive comments:

Line 17-18: should say “Bathylagidae and Myctophidae dominated the otolith record.” It is not correct or appropriate to imply that the otolith record is a quantitative measure of assemblage composition. The otolith record should only be used as a RELATIVE representation of abundance over time for only those fishes whose otoliths are amenable to preservation. For example, stomiiform fishes have very small otoliths that are likely underrepresented in the record. These fishes are the numerically dominant mesopelagic fishes in the World Ocean, and presumably were so during the timespan of this scale record.

We agree with Dr. Sutton that the otolith record cannot be assumed to be an accurate representation of the overlying fish assemblage. We have revised our entire manuscript to focus on fishes represented in the otolith record of the Santa Barbara Basin (SBB).

We also state that dominance of mesopelagic fish in the SBB otolith record is consistent with the dominance of otoliths of mesopelagic fish elsewhere in the world (lines 73-75) and with trawl data for the SBB (Brown 1974, Nishimoto and Washburn 2002) (lines 76-78). Brown (1974) studied the SBB, the Santa Cruz Basin (SCB, adjacent and SSE of the SBB) and the Rodriguez Dome Area (RDA, adjacent and WSW of the SBB). A 6-ft IKMT with a ½”-stretch mesh net and opening-closing cod end was used at night on 28 cruises in 1964-1966 resulting in 210 collections in the SBB (13,800 individuals), 267 collections in the SCB (11,415 individuals) and 91 collections in the RDA (2,999 individuals). Sampling was from surface to near bottom in the SBB ~ 540 m and to 1000 m in the SCB and RDA. The total volume of fishes, standardized by total effort, was 47.39 for the SBB, 13.17 for the SCB and 12.18 for the RDA (units: number of individuals per kilometer trawled; Brown 1974, Table 2). The total number of species captured was 44 for the SBB, 67 for the SCB and 57 for the RDA. Hence, the SBB had a relatively large and low-diversity fish assemblage.

Rather than paraphrasing, I have cut and pasted composition results below (Brown 1974, p. 17):

Composition. The faunal overlap between the two offshore areas was much greater than that of either area with the Santa Barbara Basin. Some 39% of 81 species occurred in all three areas, while 61% occurred in the Santa Cruz and Rodriguez areas only.

The seven most abundant species comprised more than 90% of the individuals in the Santa Barbara and Santa Cruz basins, but only 78% in the Rodriguez Dome area (Table 4). Only two species, the deepsea smelt *Leuroglossus stilbius* and lanternfish *Stenobranchius leucopsarus* made up over 90% of the captures in the Santa Barbara Basin. In the Santa Cruz Basin, however, they, along with the bristlemouth *Cyclothone signata* and lanternfish *Triphoturus mexicanus*, made up about equal proportions of the total fish catch: 11–17% of the total number of individuals captured, taken in 58–69% of the collections. The bathypelagic bristlemouth *Cyclothone acclinidens* was the most abundant and most frequently captured species at Santa Cruz. Two other lanternfishes, *Diaphus theta* and *Lampanyctus ritteri*, were taken less frequently in smaller numbers.

Thus, Brown (1974) shows the SBB to have an abundant, low-diversity assemblage of fishes dominated by two species, the myctophid *Stenobranchius leucopsaurus* and the bathylagid *Leuroglossus stilbius*. The stomiiforme *Cyclothone signata* was the third-most-abundant species in the SBB, comprising 4% of the total number of individuals. Conversely, *C. signata* was the most abundant species in the RDA and *C. acclinidens* was the most abundant species in the SCB (28% and 27% of the total number of individuals, respectively) (Brown 1974, Table 4). The results for the SCB and RDA are similar to those for the NE Pacific Ocean as reported by Davison *et al.* 2013 (Davison, Checkley *et al.* 2013) (the SBB was not sampled). Smaller individuals of *Cyclothone* spp. may have been extruded through the net used by Brown (1974). However, given the similarity of Brown (1974)'s results for outside the SBB (i.e., SCB and RDA) with results from the use of smaller-mesh trawls (Davison *et al.* 2013) in the same region, Brown (1974)'s results for *Cyclothone* spp. the SBB may not be biased.

We agree with Dr. Sutton and state (lines 95-97, 423-4277) that the otoliths of stomiiforme fishes, particularly *Cyclothone* spp., are small (Campana 2004), may be underrepresented in the otolith record, due to poor preservation and/or retention during our analysis. We used a 104- μ m mesh to retain particles, including otoliths, when processing our samples. Unaltered otoliths of stomiiforme fishes, other than *Cyclothone* spp., have a minor axis length large enough to ensure high or full retention (Campana 2004, (Jones and Morales 2014)). However, *Cyclothone* spp. are small and thus may not be well represented in the fossil otolith record.

(Nishimoto and Washburn 2002) report on night trawls from the upper 40 m in the SBB in June 1998 and 1999 using a 12-m x 12-m Cobb trawl with a “9-mm mesh cod end”, equivalent to an 18-mm-stretch mesh. The myctophid *S. leucopsaurus* and the bathylagid *L. stilbius* were most abundant. *Cyclothone* spp. is unlikely to occur in the upper 40 m and was not reported.

We now restrict our attention to the fossil otolith record from the SBB. We acknowledge the similarity of the mean taxonomic composition of the fossil otolith record and the taxonomic composition of trawl collections.

We also state that the SBB mesopelagic fish assemblage is relatively large and low diversity (line 153). Thus, it is similar to the global ocean in having a large mesopelagic fish assemblage but differs from the global ocean in having a low-diversity fish assemblage. These factors combined with the anoxic bottom waters of the SBB provide a unique opportunity to study the long-term variation of a mesopelagic fish assemblage by analysis of the otolith record in the sediments.

18-20: The meaning of these statements is unclear – in a global sense, gelatinous zooplankton dominate in colder waters and copepods in warmer waters. If this situation is reversed on the SBB, then this would severely limit the global application of these findings.

We agree with Dr. Sutton that our statement was inaccurate. We have deleted this section.

142-149: This section, which is one of the central foci of this paper, has major problems.

We have deleted this section.

The statement “The Bathylagid *L. stiblius* prefers gelatinous zooplankton, which filter feed on phytoplankton, including small cells characteristic of warm, oligotrophic conditions,” is in error on many counts. First, the ‘gelatinous zooplankton’ span several phyla, only one of which (Chordata, subphylum Tunicata) filter feeds small phytoplankton. The majority of gelatinous zooplankton are predaceous.

We have deleted this section.

Second, it has not been proven that this bathylagid consumes only salps. The exact diet composition of gelatinous-feeding fishes is notoriously difficult to determine (see review papers of Arai).

We have deleted this section.

Third, as general (global) rule, salps tend to dominate the zooplankton of high-nutrient oceanic regimes (esp. higher latitudes), whereas crustacean zooplankton (copepods) dominate lower-nutrient oceanic regimes (lower latitudes) – the authors have stated just the opposite.

We have deleted this section.

Thus, this thesis in the paper may be conceptually flawed, with the following points supporting this assertion: 1) the alternate stable states suggested here are unfounded - a large portion of the mesopelagic fish taxa are not represented in this otolith record [e.g. Stomiiforms, Aulopiformes, Stephanoberyciformes, which collectively largely outnumber myctophids and bathylagids combined]; 2) the salp/copepod dynamic is grossly oversimplified – variation in the relative abundances of these taxa in the Southern Ocean are highly volatile; 3) if, for the sake of argument, the salp/copepod dynamic were completely

accurate as depicted here, it would really suggest that the SBB is an aberrant marginal sea that is a poor analog for the World Ocean and 4) the thesis presented here implicitly assumes that the SBB is a closed basin with little/no exchange between the Pacific proper, - what if the fish assemblages were continuously refreshed/homogenized by extraneous larval recruitment?

We have deleted this thesis.

Notes on paragraph beginning in Line 151, listed by 'implication':

1. "this is the first documentation of long-term variability of mesopelagic biota..." YES – this is the most important aspect of this paper;
2. "climate appears to affect the mesopelagic realm, which acts as a filter for carbon from surface productive zone sequestered in the deep ocean." YES, another very important contribution of this paper, though, again, the mesopelagic zone is in no way a 'filter.'

Notes 1 and 2 are now the foci of the manuscript. We have deleted 'filter'.

3. "it is imperative to understand the dynamics of mesopelagic fish stocks, including their natural and human-caused variability, to sustainably manage their future fisheries." This statement falls under the heading, "Blinding glimpse of the obvious." It is not necessary, and is not a result from this paper. It can be deleted.

Deleted.

4. "Finally, two human activities, greenhouse gas emissions and fishing, have the potential to affect the mesopelagic biota." YES/NO. Yes to the first – it is a restatement of point number 2, so is essentially superfluous. No to the second – this paper did not investigate the effects of fishing on the mesopelagic biota, so does not belong here.

Deleted.

General comments:

The clarity of this manuscript could be much improved. The writing is choppy and discordant, with a tendency toward stream-of-consciousness (esp. Introduction). The number of technical errors (typos, misspellings, factual errors) suggest this paper was not thoroughly reviewed and proofed before submission. Supportive examples of this assessment are listed below.

We have attempted to improve the manuscript. The Introduction, in particular, has been revised (lines 9-21). The revised manuscript has been reviewed and proofed.

Throughout ms: "fishes" should be used in all cases when referring to more than one species.

Corrected.

Line 17, 46: No, not all fishes have three pairs of otoliths (e.g., Chondrichthyes).

We now use 'bony fishes' throughout the manuscript.

Teleost fishes have three pairs of otoliths (Popper, Ramcharitar *et al.* 2005). Chondrichthyes have many, small otoconia comprising each mass of the pairs of saccular and utricular otoliths (Lychakov and Rebane 2000). We did not sample otoconia; their small size makes it likely they do not preserve well and would not be retained in our analysis.

31: the mesopelagic zone does not act as a filter. This needs to be rewritten.

Deleted.

76: bony

Corrected.

83-86: fish family common names are not capitalized; cannot start sentence with genus abbreviation.

Corrected.

88: formula typographical error.

The correct unit for otolith deposition rate is otoliths $(100 \text{ cm}^2 \text{ y})^{-1}$ (line 105).

89: poor grammar – what does this sentence mean?

Reworded (lines 105-107).

97-99: this statement is vague to the point of being irrelevant.

Deleted.

129: larval

Corrected.

Final recommendation: this paper contains valuable information, but also several sections that require revision. I recommend the authors revise and resubmit, as the paper is not publishable as-is.

We appreciate this thorough and constructive review.

Signed: Tracey T. Sutton

Reviewer #2 (Remarks to the Author):

The central premise of this MS is excellent, and very appropriate for the readership of Nature. Fish otoliths provide an excellent means of species identification for some taxa, and tend to be better preserved in non-acidic sediments than are scales. Therefore the approach used in this study offers great potential. In general, the study seems to have been carefully done and well written, with one serious caveat: why are the results so different than those previously published for the same location? The previous studies by Soutar and Isaacs (1969) and Baumgartner et al (1982) used the same general approach, in the same location, using scales rather than otoliths. Although the authors' explanations for differential preservation of scales vs otoliths are valid, they do not provide a convincing case for the validity of one millennial reconstruction over the other. Therefore, I recommend that the MS be re-evaluated after a better 'reality check' is provided.

We agree with Dr. Campana that the fossil otolith and scale records must be reconciled. We now explicitly do so. We state in the first paragraph (lines 19-21) that the two records are complementary. We have added a paragraph in the main body of the manuscript directly addressing this issue (lines 81-103) and have added the section Otolith Preservation to the Methods (lines 396-427). We now focus on the otolith record per se but also state that it is consistent with mesopelagic fishes dominating trawl collections (Brown 1974, Nishimoto and Washburn 2002).

These and other suggestions are detailed below.

Line 38: This sentence needs to refer specifically to the study location, since it implies (incorrectly) that sardines and anchovy scales occur in all anoxic sediments, and that they are the only species whose scales accumulate.

Deleted.

Line 72: Although it is fair to state that some fish do not have scales, it is misleading to suggest that the mesopelagic fish reported in this study do not have scales. As a matter of fact, myctophid scales are very large, are they not?

We now state "Mesopelagic fishes dominant in trawl collections from the SBB either do not have scales (the bathylagid *L. stilbius*) or have fewer scales that are very thin and thus unlikely to preserve well (the myctophid *S. leucopsarus*)." (lines 87-89).

We are unable to identify fossil otoliths to species and hence we do not know of scale production by the dominant mesopelagic taxa in the SBB otolith record. Myctophid scales are

similar in area to anchovy, sardine and hake (Brager and Moritz 2016) but very thin (Eigenmann and Eigenmann 1890). Mesopelagic fishes have fewer vertebrae, which vary with the number of lateral line scales, than anchovy, sardine and hake and thus have fewer scales.

Line 78: The statement that degradation rate differences between scales and otoliths might explain the very different species composition apparent in this study compared to earlier scale-based studies is a reasonable one. However, I was surprised that there appears to have been no adjustment made for species-specific otolith degradation rates. It is well established that small otoliths or those with a high surface area to volume ratio dissolve or degrade more quickly than do more robust otoliths. As a result, the species with more fragile otoliths tend to be under-represented in studies such as these (i.e. Tollit et al 1997. Species and size differences in the digestion of otoliths and beaks: implications for estimates of pinniped diet composition. Can. J. Fish. Aquat. Sci. 54:105-119). That may explain the relative absence of sardine and anchovy otoliths in this study compared to past scale-based studies reporting high abundance of sardines and anchovies in the same Basin, but does not indicate that the current study provides the more accurate species reconstruction.

The Methods section Otolith Preservation (lines 396-427) addresses why a correction for interspecies variation in otolith preservation is not feasible.

The literature on corrections for otolith alteration focuses largely on otoliths recovered from feces of known predators, particularly marine mammals (Tollit, Steward *et al.* 1997, Orr and Harvey 2001, Wilson, Grellier *et al.* 2017). As we state (lines 412-415), the path hence environment of otoliths from live fishes to the sediment in the SBB is unknown.

Therefore, I strongly suggest that steps be taken to provide a reality check for the current results. The easiest way to do this would be to relate recent fish surveys of the Basin area with the otolith-based species composition inferred from the most recent period of the sediment cores, using all species, not just the summary reported in the text. Indeed, given that the authors reported no temporal trends in species composition, perhaps the entire sediment core time series could be used. Either way, it is important to demonstrate that the two species composition patterns (survey and otolith-based) correspond.

The best available fish surveys are Brown (1974) and Nishimoto and Washburn (2002). We agree that the best comparison is between the fossil otolith record from the entire sediment core and trawl data. These are consistent in showing dominance by mesopelagic fishes (lines 76-78). If we compare fossil otolith data only from 1934-2004, a period bracketing the 1964-1967 SBB sampling by Brown (1974), mesopelagic fish otoliths comprised ~ 88% of all identified otoliths (Supplementary Data Table 1) compared with 90% in Brown (1974). However, this comparison is based on only 17 otoliths.

We also cite papers on otoliths in sediments from other regions of the world. Uniformly, these show mesopelagic fishes, most often myctophids, to dominate (lines 73-75).

Given that the sediment cores reflect a seasonal integration of fish abundance, it would be best if seasonal fish surveys of the Basin were used in the otolith-fish survey comparison. However, one survey is better than nothing.

We agree that seasonally resolved data would be ideal. However, we are unaware of either recent or fossil data with seasonal resolution.

I note as well that one of the two references used in support of recent high abundance of midwater fishes was from the Santa Barbara Channel, not the Santa Barbara Basin. Are these different areas?

These are the same areas. The Santa Barbara Channel is above the Santa Barbara Basin.

If some species seem to be under-represented in the sediment cores relative to fish surveys, then some sort of calibration will be required to adjust for the species detection bias in the cores.

Please see above.

Line 107: A lot of emphasis and space is given to the power spectra, without much justification for why it is done. There is very little in the way of convincing long-term periodicity evident in any fish species abundance time series, so why would you expect it here? More importantly, where are some of the other, more convincing, statistical analyses? Where is a PCA to highlight similarities among time series? Or better yet, a mixed effects model to test for significant effects of environmental proxies on the fish abundance time series?

We agree that insufficient justification was provided for our time-series analyses. We have added a sentence (lines 123-124) and revised the presentation and treatment of the results (lines 124-134). Global spectra are now provided for ODR and environmental proxies with significance levels shown (filtered, Fig. 4; unfiltered, Extended Data Fig. 3).

Phenomena that potentially force the pelagic, including the epi- and mesopelagic, at scales detectable in the fossil otolith record include the Pacific Decadal Oscillation (50-70 y period, but shorter and longer as well) and solar activity (70-200 y period, also shorter and longer) (lines 145-151). Because our fossil otolith data come from the analysis of different cores than used to develop proxies for SST and PROD, we felt more confident in comparing properties of those time series in both the time and frequency domains than direct comparisons in time, e.g., correlations, PCA or mixed effects models. Moreover, the relatively few taxonomic groups (Fig. 2) we use and the dominance by two (Bathylagidae and Myctophidae) preclude most multivariate methods. The methods we have used to test for significance in our analyses are accepted in the literature, as cited in the Methods (lines 515-573).

Line 117: This analysis hinges on the assumption of a stable otolith degradation rate (not to be confused with deposition rate). Are there any studies which support the assumption that the anoxic conditions and pH at the bottom of the Basin remain stable across years, independent of SST or productivity? If so, these should be cited. If not, the implications of the bias need to be mentioned.

The lack of trend in ODR over two millennia for the individual families and Composite is consistent with a lack of time-dependent alteration of otoliths in the sediment (lines 105-107). The pH at and to 8 cm below the sediment surface was relatively high (7.5-8.2) in 1993 (Reimers, Ruttenberg *et al.* 1996). We are unaware of any other direct measurements of pH in the SBB sediment. Flushing of deep water in the SBB has been inferred from observations of deep water (Bograd, Schwing *et al.* 2002, Goericke, Bograd *et al.* 2015). We are unaware of time series of observations of SBB sediment characteristics (pH, O₂). Chronologies of the SBB indicate variable deposition of sediments over the past two millennia remains well established (Hendy, Dunn *et al.* 2013, Schimmelmann, Hendy *et al.* 2013). While conditions in the SBB have varied over the past two millennia, there are no data that we are aware of indicating a bias that would lead to our results. We feel our observations on the lack of trends in ODR (Fig. 2) and the literature we cite on the SBB fossil record (Baumgartner, Soutar *et al.* 1992, Kennett and Kennett 2000, Berger, Schimmelmann *et al.* 2004, Field, Baumgartner *et al.* 2006, Schimmelmann, Lange *et al.* 2006, Barron, Bukry *et al.* 2010, Hendy, Dunn *et al.* 2013, Schimmelmann, Hendy *et al.* 2013, Barron, Bukry *et al.* 2015, Skrivanek and Hendy 2015) support our use of the SBB fossil record over the past two millennia, including calcareous fossils.

Line 133: The glaring omission here, both in the text and in the Results, are the pelagic fish species, particularly sardines and saurys. Why weren't any of their otoliths found? And why were so few mesopelagic fish scales reported in the earlier studies of Soutar and Isaacs and Baumgartner? And even if some explanation is found for the huge discrepancy between the authors' results and those of scale-based studies, why is there no discussion of it (other than the brief mention in the introductory comments on Line 72)? This definitely needs to be addressed in the text. Either the previous studies were hugely biased, or the current study is hugely biased. Both approaches have identifiable sources of bias. It is important that the authors convince the reader that their study is the one that should be believed.

Please see replies above.

Line 156: Although it is reasonable to assume that greenhouse gas emissions and fishing have the potential to affect mesopelagic fish abundance, there is nothing in this study that would support that conclusion.

Deleted.

Line 424: According to Line 398, only a subset of otoliths were analyzed for elemental composition. However, it states here that the model DFA10SW (of which two of the input

variables are elemental composition) was used to identify all otoliths. One of these statements appears to be wrong.

Corrected (lines 485-487).

We appreciate this thorough and constructive review.

Steve Campana

Reviewer #3 (Remarks to the Author):

This paper reports data on the otoliths of mesopelagic fish in the Santa Barbara Basin to raise several interesting point. First, mesopelagic fish dominated the otolith assemblage in the sediments, testifying to their high abundance and likely importance in carbon cycling. Second, climatic effects may influence ecological dynamics at mesopelagic depths. Third, and related, the otoliths of mesopelagic fishes should provide clues to those impacts and potentially serve as proxies for climatic influence at those depths.

There is one potentially fatal flaw in this paper. From my reading, all otoliths of all fishes were identified and enumerated (to the extent possible, of course). The issue is the danger of pseudoreplication. It is just like collecting clam shells on the beach and counting both left and right valves. Paleontologists routinely choose one or the other to avoid pseudoreplication, and studies have shown that one valve or the other can in some situations be favored to wash up depending on hydrography. Likewise, at least potentially, for otoliths: is it possible that the three types of otoliths are not equally preserved?

We agree that pseudoreplication should be avoided. However, the condition of the fossil otoliths we recovered does not enable their classification as right or left. We use expert opinion and the otolith dimensions, shape and elemental composition to classify fossil otoliths from the SBB. The properties that enable classification of side of origin, including the sulcus, are not sufficiently well preserved to allow classification as right or left. In addition, the likelihood of both right and left sagittal otoliths of a single fish occurring in a single transverse section (our unit of sampling, each representing ~ 2-8 y, lines 384-385) of a core is small. Approximately 39% of transverse sections contained two or more otoliths of the same family, accumulated over ~ 2-8 y. The likelihood that both were from the same fish is small. Thus, pseudoreplication, while important, is unable to be corrected for and unlikely. A left or right bias in otolith preservation is also unlikely, as left-right asymmetry in otoliths is small (Lychakov and Rebane 2005).

We assume we are recovering and analyzing sagittal otoliths. As stated in the manuscript (lines 44-45), the sagittae are the largest of the three pairs of otoliths in bony fishes (Lychakov and Rebane 2000). The mass (area) of sagittae is 50-75 (14-17) times that of lapilli and asterisci. The smaller lapilli and asterisci are likely to not preserve well (Lychakov and Rebane 2000). Only sagittae are in catalogues and atlases of fish otoliths (Campana 2004, Jones and Morales 2014, Agiadi, Girone *et al.* 2018).

Could there be interspecific differences in preservation?

Yes, interspecific differences in preservation may exist. Preservation differences are likely due to differences in otolith size at the time of death, smaller otoliths being less-well preserved (please see above). Differences in chemical composition may also affect preservation, although we are unaware of data on this, particularly for the families dominating the otolith record of the SBB. While otolith preservation may vary between species, we are unable to correct for this. We do not feel it is a bias capable of explaining our results.

The authors should only be counting the largest otoliths on one side: the left or right sagittae, because the sagittae are the largest ones. It should be possible to rerun the analysis with only one of the sagittae.

Please see above.

Literature Cited

Agiadi, K., *et al.* (2018). Pleistocene marine fish invasions and paleoenvironmental reconstructions in the eastern Mediterranean. *Quaternary Science Reviews* **196**: 80-99.

Barron, J. A., *et al.* (2010). Santa Barbara Basin diatom and silicoflagellate response to global climate anomalies during the past 2200 years. *Quaternary International* **215**: 34-44.

Barron, J. A., *et al.* (2015). High-resolution paleoclimatology of the Santa Barbara Basin during the Medieval Climate Anomaly and early Little Ice Age based on diatom and silicoflagellate assemblages in Kasten core SPR0901-02KC. *Quaternary International* **387**: 13-22.

Baumgartner, T. R., *et al.* (1992). Reconstruction of the history of Pacific sardine and northern anchovy populations over the past two millennia from sediments of the Santa Barbara Basin, California. *California Cooperative Oceanic Fisheries Investigations Reports* **33**: 24-40.

Berger, W. H., *et al.* (2004). Tidal cycles in the sediments of Santa Barbara Basin. *Geology* **32**: 329-332.

Bograd, S. J., *et al.* (2002). Bottom water renewal in the Santa Barbara Basin. *Journal of Geophysical Research-Oceans* **107**(C12): 3216.

Brager, Z. and T. Moritz (2016). A scale atlas for common Mediterranean teleost fishes. *Vertebrate Zoology* **66**: 275-386.

Brown, D. W. (1974). Hydrography and midwater fishes of three contiguous oceanic areas off Santa Barbara, California. *Natural History Museum, Los Angeles, Contributions in Science*. Los Angeles, CA, Natural History Museum: 1-30.

Campana, S. E. (2004). Photographic Atlas of Fish Otoliths of the Northwest Atlantic Ocean. Ottawa, Canada: 1-284.

Davison, P. C., *et al.* (2013). Carbon export mediated by mesopelagic fishes in the northeast Pacific Ocean. *Progress in Oceanography* **116**: 14-30.

Eigenmann, C. H. and R. S. Eigenmann (1890). Additions to the fauna of San Diego. *Proceedings of the California Academy of Sciences (Series 2)* **3**: 1-24.

Field, D. B., *et al.* (2006). Planktonic foraminifera of the California Current reflect 20th-century warming. *Science* **311**: 63-66.

Goericke, R., *et al.* (2015). Denitrification and flushing of the Santa Barbara Basin bottom waters. *Deep-Sea Research Part II-Topical Studies in Oceanography* **112**: 53-60.

Hendy, I. L., *et al.* (2013). Resolving varve and radiocarbon chronology differences during the last 2000 years in the Santa Barbara Basin sedimentary record, California. *Quaternary International* **310**: 155-168.

Jones, W. A. and M. M. Morales (2014). Catalog of otoliths of select fishes from the California Current System. <http://escholarship.org/uc/item/5m69146s>.

Kennett, D. J. and J. P. Kennett (2000). Competitive and cooperative responses to climatic instability in coastal southern California. *American Antiquity* **65**: 379-395.

Lychakov, D. V. and Y. T. Rebane (2000). Otolith regularities. *Hearing Research* **143**: 83-102.

Lychakov, D. V. and Y. T. Rebane (2005). Fish otolith mass asymmetry: morphometry and influence on acoustic functionality. *Hearing Research* **201**: 55-69.

Nishimoto, M. M. and L. Washburn (2002). Patterns of coastal eddy circulation and abundance of pelagic juvenile fish in the Santa Barbara Channel, California, USA. *Marine Ecology Progress Series* **241**: 183-199.

Orr, A. J. and J. T. Harvey (2001). Quantifying errors associated with using fecal samples to determine the diet of the California sea lion (*Zalophus californianus*). *Canadian Journal of Zoology-Revue Canadienne De Zoologie* **79**: 1080-1087.

Popper, A. N., *et al.* (2005). Why otoliths? Insights from inner ear physiology and fisheries biology. *Marine and Freshwater Research* **56**(5): 497-504.

Reimers, C. E., *et al.* (1996). Porewater pH and authigenic phases formed in the uppermost sediments of the Santa Barbara Basin. *Geochimica et Cosmochimica Acta* **60**: 4037-4057.

Schimmelmann, A., *et al.* (2013). Revised ~ 2000-year chronostratigraphy of partially varved marine sediment in Santa Barbara Basin, California. *GFF* **135**: 258-264.

Schimmelmann, A., *et al.* (2006). Resources for paleoceanographic and paleoclimatic analysis: A 6,700-year stratigraphy and regional radiocarbon reservoir-age (AR) record based on varve counting and C-14-AMS dating for the Santa Barbara Basin, offshore California, USA. *Journal of Sedimentary Research* **76**: 74-80.

Skrivanek, A. and I. L. Hendy (2015). A 500 year climate catch: Pelagic fish scales and paleoproductivity in the Santa Barbara Basin from the Medieval Climate Anomaly to the Little Ice Age (AD 1000-1500). *Quaternary International* **387**: 36-45.

Tollit, D. J., *et al.* (1997). Species and size differences in the digestion of otoliths and beaks: Implications for estimates of pinniped diet composition. *Canadian Journal of Fisheries and Aquatic Sciences* **54**: 105-119.

Wilson, L. J., *et al.* (2017). Improved estimates of digestion correction factors and passage rates for harbor seal (*Phoca vitulina*) prey in the northeast Atlantic. *Marine Mammal Science* **33**: 1149-1169.

Reviewers' Comments:

Reviewer #2:

Remarks to the Author:

The authors have done a good job of addressing some of my previous concerns with this MS. Unfortunately, the conclusions were simply not that compelling; the stated link between otolith deposition rate, abundance and climate was hard to discern, and even if present, was exactly as would have been expected. Therefore, I am unable to recommend that this MS be accepted for publication in Nature Communications. Comments follow:

Lines 40-42: This sentence is a non-sequitur. The paragraph is spent discussing the importance of mesopelagic fishes, but the otolith record in the sediments reflects both the pelagic and mesopelagic community, with no way to separate them.

Lines 73-75: The global abundance of mesopelagic fishes could be discussed towards the end of the paper, but its mention here merely confuses the issue of abundance in the study area.

Line 126: Why were time series <30 yr filtered out? Because of sediment timing resolution?

Lines 123-134: The power spectra are underwhelming. None of the series show obvious coherence. I questioned the value of including these spectra in the original review, and I repeat it here.

Line 161: Is this the central conclusion of this paper? That mesopelagic fish abundance varies with climate? While there are good theoretical reasons to accept that conclusion, nothing in this paper really convinced me of that. As a matter of fact, my take-home message from this study revolved around the relative stability of the fish community over a period of 2000 years, including over the past few centuries when fishing effects might have been expected to radically disrupt relative species abundance. Yet this issue was not discussed. An equally viable conclusion is that environmental conditions, particularly SST and pH, affected otolith degradation rates, and thus the otolith deposition rate in the sediments. If so, the variability in the otolith record might not reflect species abundance at all, but otolith preservation.

This is a useful and publishable study, and one which I was hoping to recommend for publication in Nature Communications. However, other than the impressive scale of the time series, it just doesn't have the novelty or convincing conclusions that I associate with this journal.

Reviewer #3:

Remarks to the Author:

The authors have responded appropriately to my concern about pseudoreplication. I have no further issues, and from my perspective the paper is ready for publication.

Response to Reviewers' Comments

Reviewers' comments are in bold. Authors' replies are in normal font. Line numbers in replies refer to revised manuscript.

Reviewer #2 (Remarks to the Author):

The authors have done a good job of addressing some of my previous concerns with this MS. Unfortunately, the conclusions were simply not that compelling; the stated link between otolith deposition rate, abundance and climate was hard to discern, and even if present, was exactly as would have been expected. Therefore, I am unable to recommend that this MS be accepted for publication in Nature Communications. Comments follow:

Lines 40-42: This sentence is a non-sequitur. The paragraph is spent discussing the importance of mesopelagic fishes, but the otolith record in the sediments reflects both the pelagic and mesopelagic community, with no way to separate them.

We revised this sentence to state that our aim was to use otoliths to separate pelagic, mesopelagic and demersal fishes in the fossil record, i.e., otoliths provide "a way to separate them". (Lines 43-46).

Lines 73-75: The global abundance of mesopelagic fishes could be discussed towards the end of the paper, but its mention here merely confuses the issue of abundance in the study area.

These lines are in what is now named the Results and Discussion section. We feel this is the best place to discuss our result that otoliths of mesopelagic fish assemblage dominated the fossil record in the Santa Barbara Basin (SBB). Our intent is show our results for the SBB study area are consistent with those from diverse locations worldwide and thus not anomalous.

Line 126: Why were time series <30 yr filtered out? Because of sediment timing resolution?

Yes and now stated in this sentence (Lines 145-148). As stated in the original and final versions of the manuscript, both filtered (Fig. 4) and unfiltered (Supplementary Figure 3) yield similar results.

Lines 123-134: The power spectra are underwhelming. None of the series show obvious coherence. I questioned the value of including these spectra in the original review, and I repeat it here.

We have chosen to retain the global power spectra in the main body of the manuscript with additional explanation (Lines 143-158). The significant peaks we detected for both fish and climate over the range of periods for dominant climate processes is noteworthy and consistent with an effect of climate on the mesopelagic. The spectral analyses are rigorous and thus the results allow strong inference about the existence of periodicity in fish abundance and climate

proxies. Significant coherence (regardless of phase) between pairs of time series in a statistical sense is less likely to be observed than significant peaks in power spectra of single time series, nor is it necessarily expected between fish and climate when data come from different cores. Finally, the power spectra are but one type of result consistent with the influence of climate on mesopelagic fish. Importantly, it is the combined evidence in the time (discontinuities in Fig. 3 and CUSUM plots in Fig. 5) and frequency (spectra in Fig. 4 and spectra and wavelets in Supplementary Figs. 3 and 4) domains that provides the strongest inference of climate affecting mesopelagic fish in the SBB. We now state this explicitly (Lines 155-157).

Line 161: Is this the central conclusion of this paper? That mesopelagic fish abundance varies with climate? While there are good theoretical reasons to accept that conclusion, nothing in this paper really convinced me of that. As a matter of fact, my take-home message from this study revolved around the relative stability of the fish community over a period of 2000 years, including over the past few centuries when fishing effects might have been expected to radically disrupt relative species abundance. Yet this issue was not discussed. An equally viable conclusion is that environmental conditions, particularly SST and pH, affected otolith degradation rates, and thus the otolith deposition rate in the sediments. If so, the variability in the otolith record might not reflect species abundance at all, but otolith preservation.

Yes, this is the central conclusion of our paper: "...this is the first documentation of long-term natural variability of mesopelagic biota and its relation to climate." (Lines 186-187). We have shown significant variation in the otolith deposition rate (ODR) of families of fishes and their composite as well as environmental proxies using time and frequency domain analyses. We have not discussed fishing because (a) no fisheries exist for mesopelagic fish and (b) fishing effects on fishes of other families (e.g., hake, anchovy and sardine), if present, would be present during only the last ~ 5% (100 y) of our 2000 y time series and are not detectable in our data. We have addressed otolith preservation in our prior Replies to Reviewers' Comments and the final revised manuscript (Lines 254-280) and concluded it does not explain observed variations in ODR.

This is a useful and publishable study, and one which I was hoping to recommend for publication in Nature Communications. However, other than the impressive scale of the time series, it just doesn't have the novelty or convincing conclusions that I associate with this journal.

Reviewer #3 (Remarks to the Author):

The authors have responded appropriately to my concern about pseudoreplication. I have no further issues, and from my perspective the paper is ready for publication.